# Investigating the Cellular Uptake of Model Nanoplastics by Single-Cell ICP-MS

**DOI:** 10.3390/nano13030594

**Published:** 2023-02-01

**Authors:** Domenico Cassano, Alessia Bogni, Rita La Spina, Douglas Gilliland, Jessica Ponti

**Affiliations:** European Commission, Joint Research Centre (JRC), 21027 Ispra, Italy

**Keywords:** nanoplastics, cell interaction, single-cell ICP-MS, electron microscopy

## Abstract

A synthetic route to producing gold-doped environmentally relevant nanoplastics and a method for the rapid and high-throughput qualitative investigation of their cellular interactions have been developed. Polyethylene (PE) and polyvinyl chloride (PVC) nanoparticles, doped with ultrasmall gold nanoparticles, were synthesized via an oil-in-water emulsion technique as models for floating and sedimenting nanoplastics, respectively. Gold nanoparticles were chosen as a dopant as they are considered to be chemically stable, relatively easy to obtain, interference-free for elemental analysis, and suitable for bio-applications. The suitability of the doped particles for quick detection via inductively coupled plasma mass spectrometry (ICP-MS), operating in single-cell mode (scICP-MS), was demonstrated. Specifically, the method was applied to the analysis of nanoplastics in sizes ranging from 50 to 350 nm, taking advantage of the low limit of detection of single-cell ICP-MS for gold nanoparticles. As an initial proof of concept, gold-doped PVC and PE nanoplastics were employed to quantify the interaction and uptake of nanoplastics by the RAW 264.7 mouse macrophage cell line, using scICP-MS and electron microscopy. Macrophages were chosen because their natural biological functions would make them likely to internalize nanoplastics and, thus, would produce samples to verify the test methodology. Finally, the method was applied to assess the uptake by CaCo-2 human intestinal cells, this being a more relevant model for humanexposure to those nanoplastics that are potentially available in the food chain. For both case studies, two concentrations of nanoplastics were employed to simulate both standard environmental conditions and exceptional circumstances, such as pollution hotspot areas.

## 1. Introduction

Plastic pollution is acknowledged as being among the most pressing global issues and is a major threat to the environment and the welfare of the world’s wildlife. In particular, growing interest is being paid to the fraction of this debris that is in the size range below 5 mm and is commonly referred to as microplastics, the term being intended to include nanoplastics as well [1]. These fragments are ubiquitous in the environment and may originate either from the release of primary particles that were initially contained in manufactured products or as secondary particles arising from the degradation of larger plastic components. Despite the name, this definition encompasses a very wide range of sizes, down to nanosized particles that might already be present in the environment [2]. It is also worth noting that although the potential hazards from micro- or nanoplastics are particularly relevant to aquatic life, ingestion by marine species may result in the transfer of plastic debris to higher trophic levels and may possibly have an impact on human health [3]. Data generated in highly controlled conditions suggest the potential for PS-Pd nanoplastics to distribute throughout the body, translocating across the fish intestine [4].

Indeed, while most of the polymers constituting commercial plastics are chemically inert and biologically safe, many organic chemical pollutants, such as polycyclic aromatic hydrocarbons (PAHs), have a strong affinity for hydrophobic materials, raising concerns that micro- and nanoplastics could act as the carriers and concentrators of harmful contaminants [5].

Unfortunately, due to the current limitations of the most common analytical techniques used to monitor and quantify the presence of microplastics in real environment sediments, the extent of microplastic pollution has yet to be accurately determined. Moreover, evaluating the biological risk connected to the transfer of microplastics from the food chain would require the systematic analysis of complex interconnected biological models to achieve the level of statistical significance that only a high-throughput analytical technique may afford. 

In this regard, it has been recently proposed that researchers should make use of single-particle inductively coupled plasma-mass spectrometry (spICP-MS) to detect microplastics particles by monitoring the ^13^C isotope [6]. By applying this technique, microplastics could be detected in a rapid and high-throughput manner that could achieve the statistical significance needed for biodistribution studies. In addition, spICP-MS has been already exploited in other works to quantify the trophic transfer rates of nanomaterials within complex organisms and is, by now, an established technique [7,8]. However, due to the high natural background signal of carbon (as CO_2_) in the gas plasma and its poor ionization efficiency, the current limit of detection of spICP-MS for microplastics is relatively poor, being around 1.2 µm in diameter for polystyrene particles when ^13^C is chosen as an analyte. While these results represent a potentially important advance in the context of detecting micron-sized plastic particulates, they cannot contribute substantially to increasing our quantitative knowledge about the spreading of sub-micrometer plastics throughout the environment.

Since the aim of the present work was to develop an analytical procedure for the detection of statistically significant populations of plastic particles below 1 µm, we produced nanoplastics made of the environmentally relevant polymers, polyethylene (PE) and polyvinyl chloride (PVC), which were doped ad hoc with gold ultrasmall nanoparticles. Gold nanoparticles were chosen as a dopant as they are considered to be chemically stable, relatively easy to obtain, interference-free for elemental analysis, and suitable for bio-applications [9,10,11]. Indeed, gold is one of the most interesting elements for ICP-MS detection, as its absolute limit of detection is at the attogram level, meaning that even individual nanoparticles with a size below 10 nm could be detected when working in single-particle mode [12]. In practice, this allows the tracing of PE and PVC nanoplastics with a size as small as 50 nm, while maintaining a low doping level of ultrasmall gold nanoparticles. Other groups have investigated the feasibility of using metal tags as proxies for the detection of micro- and nanoplastics; however, most often, these markers are only bound to the surface of nanoplastics and might suffer from leakage [13,14,15]. Our approach supports the traceability of nanoplastics by ICP-MS, while at the same time minimizing the possibility of metal leakage and preventing the substantial alteration of the density of nanoplastics, which could have a major impact on their buoyancy, and other important properties (e.g., a chemical fingerprint, refractive index, and extinction coefficient) that could hinder their detection through other analytical techniques. 

Finally, we explored the applicability of a more recent development of the ICP-MS technique, namely, single-cell ICP-MS (scICP-MS), to the study of nanoplastics uptake by relevant biological models, such as cultured human intestinal cells. Indeed, with scICP-MS, the elemental composition of a large population of cells is evaluated at a single cell level, meaning that cells dispersed in culture media and adequately diluted are individually transported to the ICP-MS in nebulized droplets that are directly introduced into the plasma. Each droplet generates an ion cloud, which is detected as an individual current spike signal, the intensity of which is proportional to the quantity of the analyte element present in the single cell. As the number of spikes correlates with the number of cells containing the analyte of interest, it is, in principle, possible to estimate the cellular interaction of gold-doped nanoplastics for each cell. It is worth noting that scICP-MS has already been applied in single-cell analysis to quantify the cellular uptake of inorganic nanomaterials or to evaluate the cellular distribution of inorganic chemotherapeutics; hence, its potential is now acknowledged [16,17]. For this work, we also exploited the resolution and accuracy of electron microscopy to validate the results of rapid and high-throughput analysis by scICP-MS and to confirm the biodistribution.

## 2. Materials and Methods

### 2.1. Synthesis of Gold-Doped Nanoplastics

Polyethylene and polyvinyl chloride nanoparticles doped with ultrasmall gold nanoparticles (PE Au and PVC Au, respectively) in the size range of 50–350 nm were produced through a standard oil-in-water emulsion protocol. Briefly, PE (Mw ~4 KDa) or PVC (Mw ~43 KDa powders (Sigma Aldrich, Rome, Italy) were added to 3 g of toluene, which was heated until the powders had completely dissolved. Ultrasmall gold nanoparticles (diameter ≈ 3 nm) dispersed in toluene were synthesized by the standard two-phase liquid–liquid Brust method [18]. These were mixed with the polymer solution before adding 27 mL of a boiling water solution of sodium cholate (7.5 mg), which served to create a two-phase system. The two-phase system was then homogenized by an Ultra-Turrax T25 (IKA-Werke, Berlin, Germany) for 2 min at (16,000 rpm) and successively ultrasonicated with a probe ultrasonicator for 2.5 min at 40% amplitude (Vibra-Cell Ultrasonic Liquid Processors, vCX 130, Sonics & Materials, Inc., Danbury, CT, USA). After ultrasonication, the emulsion was immediately cooled in an ice-water bath for 3 min. The resulting product was passed through 5 μm polyethersulfone membrane syringe filters to remove the agglomerated/non-emulsified nanoplastics. The filtrate solution was reduced to 5 mL via rotary evaporation, then the nanoplastics were dispersed in deionized water to a final concentration of 2 mg mL^−1^ and stored at 4 °C. 

### 2.2. Cell Culture

Cell lines, Caco-2 (Sigma Aldrich, Rome, Italy) and RAW 264.7 (ATCC, Rome, Italy) were cultured under standard cell-culture conditions in a humidified incubator (37 °C, 5% CO_2_, and 95% humidity), in DMEM supplemented with 10% heat-inactivated fetal bovine serum and 1% penicillin/streptomycin (all reagents from Gibco, Rome, Italy). For exposure testing, cells were seeded in 10 cm dishes, for TEM and ICP-MS sample preparations, in 5 mL of culture medium, and for SEM sample preparation, in 6-centimeter dishes (Thermofisher, Rome, Italy) in 3 mL of culture medium containing a piece of silicon wafer on the bottom. The seeding density used was 100,000 cells mL^−1^ and cells were allowed to attach for 24 h. Finally, PE-Au or PVC-Au NPs at concentrations of 1 and 100 µg mL^−1^ were added, and exposure was allowed to continue for 48 h.

### 2.3. Sample Preparation for Single-Cell Inductively Coupled Plasma Mass Spectrometry (scICP-MS)

A Perkin Elmer NexIon 300D quadrupole ICP-MS, equipped with an SC-FAST peristaltic pump, a Meinhard concentric nebulizer, an Asperon single-cell spray chamber, and a standard quartz torch (2.5 mm internal diameter) operating in single-cell mode was used for scICP-MS analysis (Perkin Elmer, Waltham, MA, USA). For the setting of all parameters and data acquisition, the Nano Application module of the Syngistix software was used. The dwell time was set at 100 μs and the total data acquisition time was 100 s. Transport efficiency was evaluated by employing 60 nm of gold nanoparticle suspension, purchased from NanoComposix (San Diego, CA, USA) with a concentration of approximately 100,000 particles mL^–1^. The transport efficiency of the Asperon varied between 30% and 50% for the 60 nm gold nanoparticles. All data acquisition was accomplished with either the Syngistix Single-Cell or the Nano Application modules.

A critical parameter to be considered in the scICP-MS procedure is the gas flow rate. In particular, a single-cell apparatus not only needs two separate gas flows to drive the cells in the center of the linear Asperon chamber but must also maintain a low gas flow rate, in order not to burst the cells due to the high backpressure at the tip of the nebulizer. However, as the gas also has the function of cooling the plasma torch, a low flow rate increases the risk of the torch melting. For all the experiments, an optimal compromise was found to be 0.37 L min^−1^ for the nebulizer gas flow and 0.68 L min^−1^ for the auxiliary gas flow, leading to a nebulizer backpressure of about 30 psi. These conditions were found to be sufficient to minimize the risk of cells bursting while protecting against the melting of the torch. 

Another important parameter to keep under consideration in the single-cell mode is the size of the cells, which could reach several micrometers. This implies that, to avoid the overcrowding and clogging of the tubing or the nebulizer, the concentration of the cells should be much lower than 100,000 cells mL^−1^, which is the concentration of nanoparticles that has been used to verify the method. The optimal cell concentration depends on cell size; in any case, it is strongly recommended to keep the sample vial agitated to prevent any local increase in the cells’ density due to sedimentation. 

The standard calibration curves for dissolved ionic Au were prepared by diluting standard stock solutions with MilliQ water. The nanoplastics suspension samples were diluted with ultrapure water to achieve an approximate injection rate of about 1000 particles per minute. All samples were prepared in triplicate. The reported values are the average results of the three measurements.

For single-cell analyses after exposure, the cells were detached and washed in PBS until the washing solution showed no trace of gold nanoparticles when analyzed with an ICP-MS operating in single-cell mode. The appropriate concentration for scICP-MS operation was then found by analyzing increasingly dilute samples until the equivalent size distribution was found to be constant and only the total cell count varied according to concentration. This procedure was employed to minimize the possibility that more than one cell would enter the plasma during the selected dwell time of the ICP-MS detector.

In order to exclude the false positives that could arise from background noise, the single-cell application software automatically detects a threshold over which the pulses count as events. This method is valuable for the aim of this work, which is focused on a qualitative investigation of cellular interaction with nanoplastics. Nevertheless, a more rigorous analysis could be performed using the 5σ criterion, i.e., by analyzing the background noise and excluding all pulses with an intensity below a threshold defined as the mean value of the background plus 5 standard deviations [19].

### 2.4. Sample Preparation for Electron Microscopy Analysis

The NPs cell–membrane interaction and uptake were qualitatively studied via electron microscopy analyses. After exposure, cells were washed twice with PBS and then prepared for SEM and TEM analysis, as follows. All chemicals, unless otherwise stated, were purchased from Sigma Aldrich (Rome, Italy).

For the TEM analysis, cells were washed, detached, and re-suspended in a 2% Karnovsky solution composed by paraformaldehyde and glutaraldehyde. Before analysis, the Karnovsky solution was removed and cells were re-suspended in an osmium tetroxide solution with 0.1 M cacodylate at pH 7.3 for 1 h. After three washes in 0.05 M cacodylate for 10 min each, cells were dehydrated in a series of ethanol solutions in MilliQ water (30%; 50%; 75%; 95% for 15 min each, and 100% for 30 min), then incubated in absolute propylene oxide for 20 min and embedded in a solution of 1:1 epoxy resinpropylene oxide for 90 min. This mixture was renewed with pure epoxy resin overnight at room temperature and was later polymerized at 60 °C for 48 h. Ultrathin sections (50–70 nm) were obtained using a Leica EM UC7 ultramicrotome (Leica, Rome, Italy) and stained for 2 min with uranyl-less solution (TAAB Laboratories Equipment Ltd., UK) and lead citrate Reynolds solution (TAAB Laboratories Equipment Ltd., UK) for 2 min, then washed and dried. Ultrathin sections were collected on Formvar carbon-coated 200 mesh copper grids (Agar Scientific, London, UK and imaged with a JEOL JEM-2100 HR-transmission electron microscope at 120 kV (JEOL, Rome, Italy). 

For SEM, the silicon wafer with cells on the surface was fixed in glutaraldehyde 2.5% in Na-cacodylate buffer 0.1 M overnight at 4 °C. The silicon wafer was then washed 3 times in Na-cacodylate buffer, 0.1 M of 5 min each, then the cells were dehydrated in a series of ethanol solutions in MilliQ water (25%; 50%; 75%; 90% for 15 min each, and twice in 100% ethanol for 15 min), then incubated for 30 min in a solution of 1:1 ethanol:hexamethyldisilazane, and finally twice in hexamethyldisilazane at 100% for 15 min. The silicon wafer was allowed to dry under a chemical hood overnight, then mounted on a stub with a biadhesive. The wafer was carbon-sputtered using a Leica EM ACE200 carbon thread coater (Leica, Rome, Italy), set on 12 pulses. The samples were imaged using SEM with an FEI NOVA 600 Dual Beam (ThermoFisher, Delft, The Netherlands).

### 2.5. Raman Spectroscopy

Samples were prepared by a 1:10 dilution of the stock suspension in water, and successive filtration on a 25 mm Anodisc (Whatman International Ltd., London, UK) filter (0.1 μm pore size). The filtrate was then washed in water and ethanol to remove the surfactant and promote the aggregation of NPs. A WITec alpha 300 (WITec, GmbH, Berlin, Germany) Raman microscope, equipped with a 532 nm laser was used to analyze the particles and produce Raman spectra, which confirmed the chemical composition of PE NPs and PVC NPs. Spectra were collected using the 100× objectives by averaging at least 20 spectra. The identification of the spectra was achieved by comparing the PE NPs and PVC NPs with the respective starting polymer powders.

### 2.6. Dynamic Light Scattering (DLS)

DLS size measurements were performed using a Malvern Zetasizer Nano-ZS instrument (Malvern Panalytical Ltd, Malvern, UK), equipped with a 633 nm HeNe laser. For each sample, a 10 μL aliquot of the 2 mg mL^–1^ mother solution was diluted with 1 mL of MilliQ water. The measurements were carried out at 25 °C and the size distribution results were the average of three consecutive measurements. All the reported graphs display intensity-based size distributions.

## 3. Results

### 3.1. Characterization of Au-Doped Nanoplastics

Hereafter, we report the physicochemical characterization on Au-doped PE and PVC nanoplastics in a size range below 1 µm. 

Electron microscopy images of PE Au and PVC Au (Figure 1a and Figure 1b, respectively) show that the nanoplastics have a spherical morphology and their size is far below the micrometer range. The size distribution histograms of PE Au and PVC Au, obtained by manually measuring the nanoplastics diameter from the SEM images, show that both types of samples have reasonably narrow size distributions, lying within the range of 50–350 nm and with a maximum of around 120 nm. TEM insets show the presence of darker spots and agglomerates that are clearly visible in the polymeric particles, confirming the successful doping with ultrasmall gold nanoparticles, which are distributed throughout the whole volume of the nanoplastics. 

Dynamic light scattering (DLS) measurements of PE Au and PVC Au in MilliQ water are reported in Figure 1c and show that both samples display the same intensity-based hydrodynamic diameters of about 200 nm. Repeated DLS measurements made over at least 3 months showed no significant change in size, confirming that under the storage conditions (MilliQ water at 4 °C) used, sodium cholate is effective in maintaining colloidal stability. 

Raman analysis was performed on both types of nanoplastics, which were previously collected by filtering with an anodisc filter of 100 nm pore size, then washed with water and ethanol to remove the excess surfactant and induce aggregation of the nanoplastics. The chemical fingerprint of the samples was obtained and a direct comparison was made between the Raman spectra of the PE Au and PVC Au samples, and the original polyethylene and polyvinyl chloride powders, respectively. Figure 1d shows Raman spectra confirming that the fingerprint regions of the nanoplastics samples correspond closely with the respective starting bulk polymer samples. In particular, the peaks in the spectral windows, spreading around 650 cm^−1^ (green) and 2930 cm^−1^ (pink), highlighted in the graph in Figure 1d, uniquely identify the polymers, which are clearly visible in both the gold-doped nanoplastics and do not overlap with the Raman spectrum of sodium cholate. It is worth noting that the presence of gold does not substantially modify the Raman spectra of polyethylene and polyvinyl chloride as the size of gold nanoparticles is below 3 nm; thus, the size-dependent radiation-damping effect prevents any sharp plasmon resonance, which might interfere with the spectra collection after laser irradiation. 

### 3.2. Single-CELL Inductively Coupled Plasma Mass Spectrometry (scICP-MS)

Figure 2 shows the equivalent sphere diameter distributions of PE Au and PVC Au, obtained through ICP-MS analysis in single-cell operating mode. As shown in Appendix A in the Appendix A, counts below 20 nm of equivalent diameter are also detected in the blanks; hence, these values could be used qualitatively to identify the minimum detectable diameter. Both the distributions are asymmetric with respect to the maximum frequency, and appear broad, with mean diameters and standard deviations of 26.1 ± 6.4 nm for PE Au (Figure 2a) and 24.8 ± 5.8 nm for PVC Au (Figure 2b). The histograms were built on a total of *N* = 1471 and *N* = 1649 counts for PE Au and PVC Au, respectively. 

### 3.3. PE and PVC Au Nanoplastics Interaction with Cells

Based on the observation that the loading of Au is adequate to ensure the detectability of single nanoplastic particles, it has been possible to move forward with exposure case studies to determine the cellular interaction of PE Au and PVC Au nanoplastics in the relevant biological models by means of scICP-MS at different exposure concentrations (1 or 100 µg mL^−1^); Table 1 summarizes the results that were obtained by each analytical technique and will be described in subsequent sections.

Figure 3 shows the results obtained from the RAW 264.7 cells after 48 h of exposure to PE or PVC Au nanoplastics at a concentration of 1 µg mL^−1^.

The equivalent sphere-size distribution of RAW 264.7 cells exposed to PE Au at the optimal cellular concentration, found through sequential dilutions, is shown in Figure 3a (equivalently, Au mass distribution is displayed in Appendix A of the Appendix A). It can be seen that the cells display a gold-equivalent size distribution, which is shifted toward higher diameters with respect to the initial single nanoplastics. This suggests that the macrophages have internalized multiple PE Au particles rather than single nanoplastics. An electron microscopy investigation of the RAW 264.7 cells belonging to the sample that was exposed to PE Au, when analyzed via scICP-MS, confirmed the uptake of nanoplastics. Indeed, the SEM images of RAW 264.7 cells (Figure 3a_1_) do not reveal any traces of PE Au attached to the external cell membrane, nor do they appear as free background materials. The TEM images of cell slices embedded in resins (Figure 3a_2_) highlight the presence of nanoplastics, which are identifiable as circular spots inside the amphisomes, marked with an A. 

Similarly, the same assessment has also been performed on PVC Au nanoplastics, with the same exposure conditions being adopted for PE Au. The scICP-MS analysis at this step revealed that while a major fraction of the cells’ population is identified with an equivalent sphere diameter of around 40 nm, meaning that the PVC Au nanoplastics have mostly been internalized as multiple particles, a non-negligible part of the population overlaps with the main peaks of the single nanoplastics (Figure 3b). In terms of the electron microscopy analysis, the results were similar to those observed for PE Au; no particles were found outside the cells, nor were they interacting with the cell membrane (Figure 3b_1_), but uptake was confirmed and the PVC Au particles were mainly found inside the endosomes (Figure 3b_2_).

The same exposure protocol was applied to the human intestinal epithelial CaCo-2 cells. Figure 4a shows the equivalent sphere size distribution of CaCo-2 cells exposed to PE Au (in parallel, Au mass distribution is displayed in Appendix A of the Appendix A). The equivalent size distribution is shifted to higher diameters with respect to PE Au distribution, again suggesting that the cells tend to be associated with more than one nanoplastic at a time. The SEM micrograph (Figure 4a_1_) of CaCo-2 cells shows a cell membrane interaction with PE Au, yet there is no evidence of uptake. The result is confirmed by the TEM analysis of CaCo-2 cell slices, where the particles are attached to the cell membrane (Figure 4a_2_).

Different results were observed when the CaCo-2 cells were incubated with PVC Au since no evidence of PVC Au uptake or transport by CaCo-2 cells was found. Electron microscopy analyses also support this argument, with no trace of nanoplastics observed either inside or outside the CaCo-2 cells during the SEM/TEM imaging (Figure 4b_1_,b_2_). 

Since the aim of this work is to assess the feasibility of employing scICP-MS as a high-throughput screening technique for environmental pollutants, we also explored a case study in which RAW 264.7 and CaCo-2 cells were again exposed to concentration of 100 µg mL^−1^. 

Figure 5 summarizes the results of the scICP-MS analysis and electron microscopy investigation of RAW 264.7 cells incubated with PE Au and PVC Au at a concentration of 100 µg mL^−1^ for 48 h (equivalently, Au mass distribution is displayed in Appendix A). Considering the equivalent sphere size distribution of RAW 264.7 cells incubated with PE Au, there is a clear shift and broadening of the distribution toward higher diameters when compared to that of the starting PE Au (Figure 5a). This might be indicative of either an increased uptake or aggregation phenomena. SEM (Figure 5a_1_,5a_2_) and TEM (Figure 5a_3_,5a_4_) investigations confirmed that while the uptake of more than one particle could justify the shift in the distribution with respect to the single-cell spectrum, it does not reflect the substantial broadening process that occurred. In fact, the scICP-MS analysis is more consistent with the presence of aggregates of tens of particles, as highlighted in Figure 5a_2_. On the other hand, PVC Au appears to behave differently from PE Au, even at high nanoplastics concentrations. Indeed, Figure 5b shows a clear shift of the distribution of RAW 264.7 cells, compared to the single-cell spectrum of PVC Au; however, the substantial broadening toward high diameters is not present and the absence of aggregated particles (Figure 5b_1_,5b_2_) in SEM investigation is consistent with the scICP-MS results. It should be noted that by increasing the concentration of nanoplastics by two orders of magnitude, we observe PVC Au cell membrane interaction (Figure 5b_3_) and the uptake has also increased, as can be seen in Figure 5b_4_, where multiple particles have been spotted inside the endosomes.

The same experiment has also been carried out with CaCo-2 cells, which were incubated with PE Au and PVC Au at a concentration of 100 µg mL^−1^ and then analyzed by both scICP-MS and electron microscopy. Similarly to the case of the RAW 264.7 cells, exposing the CaCo-2 cells to a concentration of nanoplastics that was two orders of magnitude higher contributed to an increase in the rate of uptake, which was absent when the lower, environmentally relevant concentrations were employed. In this case study, the scICP-MS analyses and electron microscopy observations were also in good agreement. Indeed, Figure 6a shows the scICP-MS distribution of PE Au and CaCo-2 cells incubated with PE Au, which display a partial superposition, meaning that most of the events being detected refer to single-particle occurrences (equivalently, Au mass distribution is displayed in Appendix A in the Appendix A). This is confirmed by SEM and TEM imaging, which show single particles present, without any evidence of aggregation (Figure 6a_1_,a_2_), whereas the uptake took place either as single or multiple particles in amphisomes (Figure 6a_3_,a_4_). A different behavior was observed for PVC Au, which showed a tendency to aggregation that was revealed both from scICP-MS (Figure 6b) and electron microscopy analysis. In the case of the scICP-MS, a shift and a broadening to high diameters of the equivalent sphere size distribution were observed, while electron microscopy imaging revealed the presence of large aggregates (Figure 6b_1_,6b_2_), cell membrane interaction (Figure 6b_3_) and multiple uptakes by CaCo-2 cells (Figure 6b_4_). 

## 4. Discussion

With the aim of developing an analytical procedure to detect plastic particles below 1 µm by ICP-MS, we have manufactured PE and PVC nanoplastics that were deliberately doped with ultra-small gold nanoparticles. Both polyethylene nanoparticles (PE Au) and polyvinyl chloride nanoparticles (PVC Au) in the size range of 50–350 nm were produced through a standard oil-in-water emulsion protocol, adding ultra-small gold nanoparticles with a mean size of 3 nm.

The choice of using gold nanoparticles in the ultra-small size range stems from the need to have uniform doping of the nanoplastics and to maximize the detection efficiency in ICP-MS. In particular, ICP-MS has a high sensitivity for gold, while the small particle size ensures that when operating in single-cell mode, there is a high probability that they will be fully ionized in the plasma [20]. To assess whether the doping level of PE Au and PVC Au was sufficient to use them as traceable model nanoplastics, scICP-MS analysis of both sample types was performed. The optimal working concentration for scICP-MS, i.e., the concentration at which particles effectively reach the torch one at a time, has been found through sequential dilutions of the mother solution (2 mg mL^−1^ dry mass basis) until the calculated mean equivalent sphere diameter was constant, while the number of detected particles decreased linearly with dilution. 

The relatively broad distribution of the equivalent size can be understood by considering the fact that the nanoplastic diameters are spread across the range of 50–350 nm, as determined by electron microscopy. Considering a reasonably uniform distribution of gold nanoparticles throughout the volume of the nanoplastics, it is clear that their equivalent sphere diameter can vary substantially and may, thus, display a broad distribution. Nonetheless, one of the possible risks related to the choice of ultra-small nanoparticles is the possibility that the metal content of the nanoplastics might not be sufficient to produce a detectable signal using the scICP-MS technique; our results, conversely, showed that the loading was adequate to ensure the detectability of single nanoplastic particles.

Based on this evidence, it has been possible to move forward with exposure case studies to determine the cellular interaction of PE Au and PVC Au nanoplastics in relevant biological models by means of scICP-MS. The first step in the feasibility study was to expose RAW 264.7 cells to both low and higher concentrations of nanoplastics; being macrophages, these are prone to internalize foreign materials, due to their intrinsic biological function, and the nanoplastics internalization was confirmed by both scICP-MS and electron microscopy techniques. 

After having assessed the feasibility of employing scICP-MS to quantify the uptake of nanoplastics by RAW 264.7 cells, we applied the same protocol to the biologically relevant human intestinal epithelial CaCo-2 cells. Considering that one of the potential routes for human exposure to nanoplastics is either via direct ingestion from drinking water or due to transfer from lower trophic organisms in the food chain, which is more likely, the use of intestinal cells to evaluate the threat represented from micro- and nanoplastics pollution is a valid choice. 

For a lower exposure concentration, we observed differences between the RAW 264.7 and Caco-2 cells. In fact, while RAW 264.7 cells were able to internalize both the PE Au and PVC Au nanoplastics, in the case of Caco-2, we have not detected any internalized PE Au or PVC Au nanoplastics; only an external cell interaction of PE Au was found, none for PVC Au, since the equivalent sphere size distribution of cells is comparable to the background noise of the instrument. Taken together, these results suggest that intestinal cells behave differently when exposed to PVC or PE nanoplastics. Considering that, overall, the mean size and the gold loading of the two samples is comparable, the changes might be due to the different chemical composition and/or density of the two materials. 

Other groups reported similar findings, showing that microplastics of different materials at selected exposure concentrations are not taken up intracellularly by CaCo-2 cells, yet they interacted with their microvilli [21] and could be detached during sample preparations for scICP-MS analysis.

These results could be of interest if we consider the external concentration of 1 µg mL^−1^ to be relevant for potential human exposure to nanoplastics, representing a reasonable compromise between the detection efficacy of scICP-MS and the actual environmental concentration of plastic debris of a size below 0.5 µm, which might be several orders of magnitude smaller than the one considered here (about 1 µg L^−1^) [22].

When cells were exposed to 100 µg mL^−1^ of PE Au or PVC Au nanoplastics, both the RAW 264.7 and Caco-2 cells showed cell interaction and internalization, detected by each technique that was applied. In particular, nanoplastics were mainly distributed inside endosomes, with amphisomes, confirming the endo-phagocytic pathway of plastics [23] and outside interactions with the cell membrane and microvilli.

It was interesting to note that the equivalent size distribution of RAW 264.7 cells, exposed to both low and high nanoplastics concentrations, was shifted to higher diameters with respect to PE Au or PVC Au distribution, suggesting that the cells tend to be associated with more than one nanoplastic at a time. 

The exposure of cells or microorganisms to realistic concentrations of nanoplastics is particularly important in studies aimed at evaluating the effect of these emerging pollutants on biological activities, such as survival, the presence of oxidative stress, feeding difficulties, or any other relevant impairment. Non-realistic dose conditions, indeed, might lead to misleading conclusions about the impact of plastic litter on the marine ecosystem.

Our results confirmed that the scICP-MS technique might be exploited to detect uptake, even at environmentally relevant concentrations of micro- and nanoplastics, or at several orders of magnitude higher than the typical concentrations found in the environment; nonetheless, they more closely match the microplastics concentration in more extreme conditions, such as those in marine sediments or water in pollution hotspot areas [24].

Indeed, with this time-resolved analysis method, plastic debris can be detected in a rapid and high-throughput manner, permitting huge numbers of cells to be analyzed in a short time and, thus, generate data, with the statistical significance needed to conduct bio-distribution assessments. Overall, we have demonstrated that our model nanoplastics can be doped with moieties of interest to enhance their traceability and, in particular, doping with ultrasmall gold nanoparticles enables their detection inside relevant biological models by means of scICP-MS.

## Figures and Tables

**Figure 1 nanomaterials-13-00594-f001:**
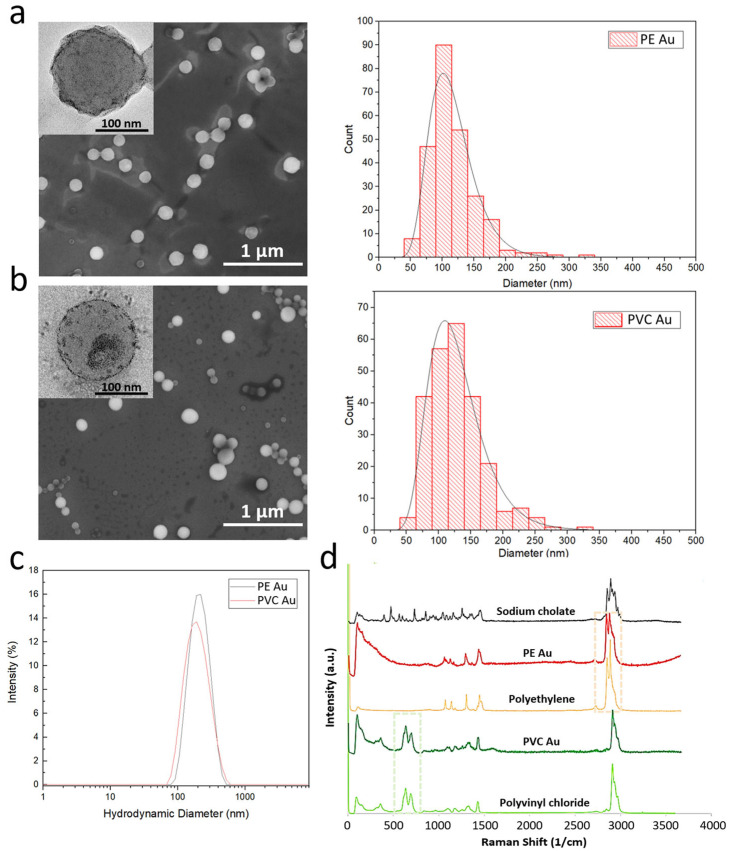
Scanning electron microscopy images of gold-doped polyethylene and polyvinyl chloride nanoplastics (images (**a**,**b**), respectively) and their mean diameter distribution, evaluated by the manual measurement of a diameter of at least 200 particles. The insets show a high-magnification transmission electron microscopy image of the single nanoplastics, in which the ultrasmall gold nanoparticles are visible as small dark spots. (**c**) DLS size intensity spectra of PE Au (blue line) and PVC Au (red line) samples, dispersed in MilliQ water. (**d**) Raman spectra of polypropylene and polyvinyl chloride polymers powders, sodium cholate hydrate powder, and PE Au and PVC Au samples.

**Figure 2 nanomaterials-13-00594-f002:**
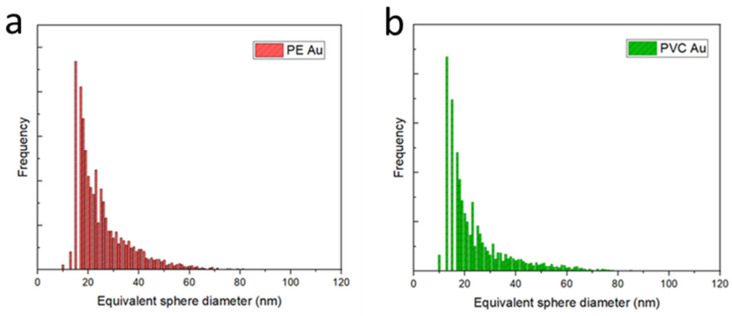
(**a**) Equivalent sphere diameter histogram distributions of gold-doped polyethylene (PE Au) and (**b**) polyvinyl chloride (PVC Au) nanoplastics, acquired in the single-cell mode of the ICP-MS.

**Figure 3 nanomaterials-13-00594-f003:**
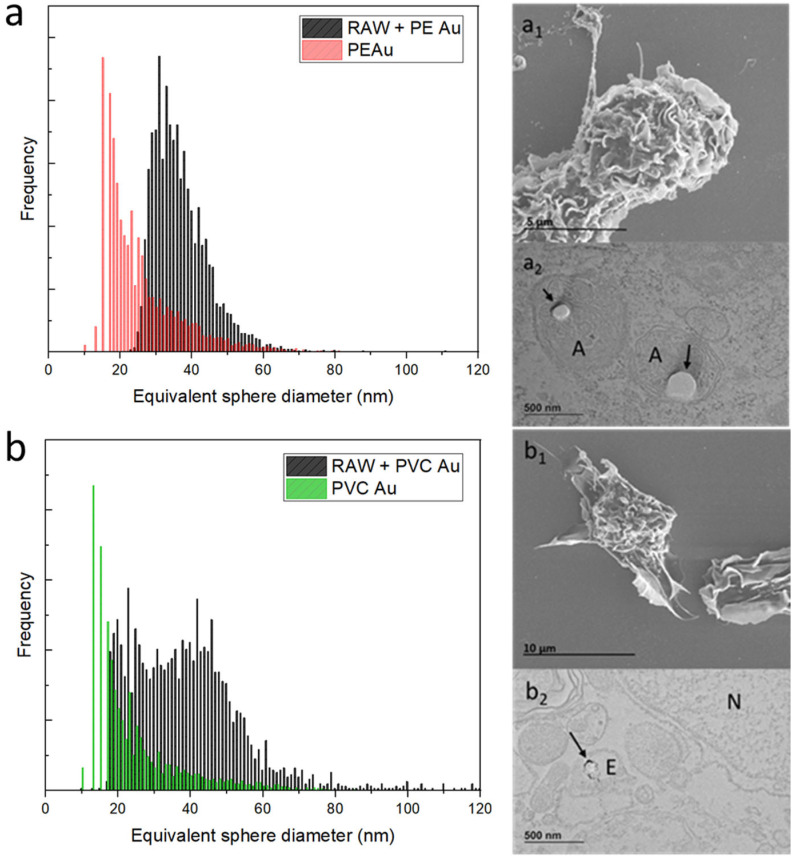
(**a**) Equivalent sphere diameter distribution of PE Au (red) and RAW 264.7 cells, incubated with PE Au for 48 h (black) at a concentration of 1 µg mL^−1^. (**a_1_**) The SEM image of the RAW 264.7 cells incubated with PE Au for 48 h does not show the presence of any nanoplastics interacting with the cell membrane or as background materials. (**a_2_**) TEM image of slices of RAW 264.7 cells after 48 h of incubation with PE Au, displaying the presence of PE Au (arrows) inside amphisomes (A), confirming the internalization. (**b**) Equivalent sphere diameter distribution of PVC Au (green) and RAW 264.7 cells incubated with PVC Au for 48 h (black) at a concentration of 1 µg mL^−1^. (**b_1_**) The SEM image of the RAW 264.7 cells, incubated with PVC Au for 48 h, does not show the presence of any nanoplastics interacting with the cell membrane or as background materials. (**b_2_**) The TEM analysis of the sample reveals the presence of nanoplastics (indicated by arrows) internalized inside the endosome (E); the nucleus is marked N.

**Figure 4 nanomaterials-13-00594-f004:**
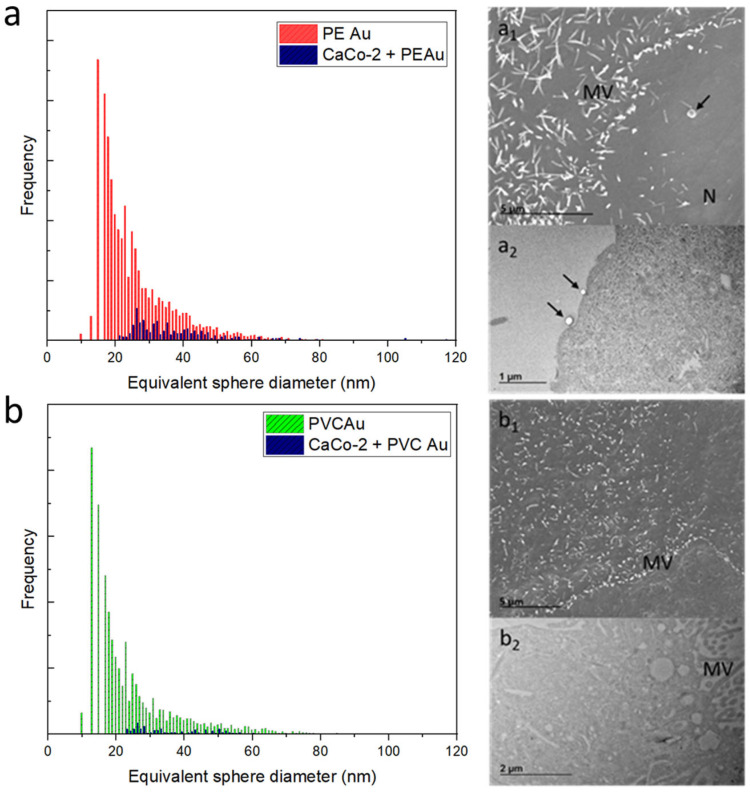
(**a**) Equivalent sphere diameter distribution of PE Au (red) and CaCo-2 cells incubated with PE Au for 48 h (blue) at a concentration of 1 µg mL^−1^. (**a_1_**) The SEM image of CaCo-2 cells incubated with PE Au for 48 h does not show the presence of any nanoplastics inside the cells, yet some nanoplastics (arrow) are clearly interacting with the cell membrane andmicrovilli (MV); nucleus (N). (**a_2_**) TEM image of slices of CaCo-2 cells after 48 h of incubation with PE Au, displaying the presence of PE Au (arrows) interacting with the cellular membrane, confirming the results of the SEM analysis. (**b**) Equivalent sphere diameter distribution of PVC Au (green) and CaCo-2 cells incubated with PVC Au for 48 h (blue) at the concentration of 1 µg mL^−1^. SEM (**b_1_**) and TEM (**b_2_**) analysis of the sample did not reveal the presence of nanoplastics, either internalized within the cells or externally interacting with the cell membrane or as a free background material.

**Figure 5 nanomaterials-13-00594-f005:**
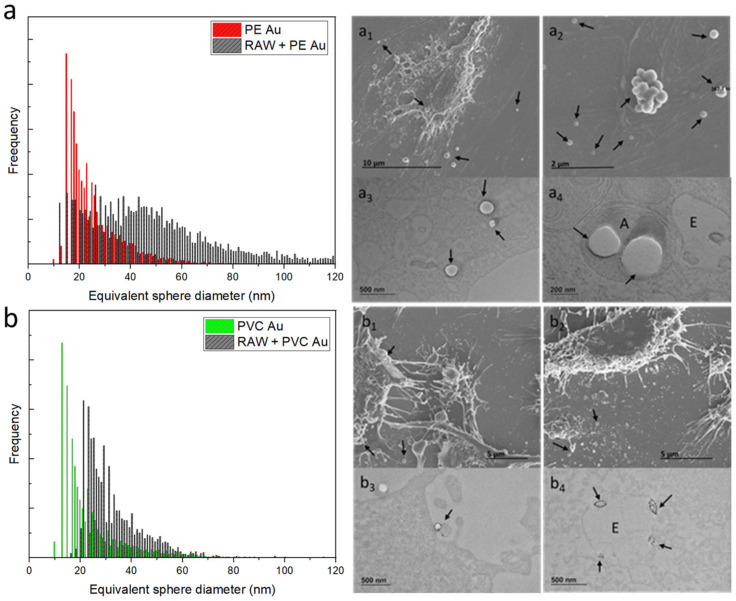
(**a**) Equivalent sphere diameter distribution of PE Au (red) and RAW 264.7 cells incubated with PE Au for 48 h (black) at a concentration of 100 µg mL^−1^. (**a_1_**,**a_2_**) The SEM image of the RAW 264.7 cells, incubated with PE Au for 48 h, reveals the presence of nanoplastics, both interacting with the cell membrane and as background materials. (**a_3_**,**a_4_**) The TEM image of the slices of RAW 264.7 cells after 48 h of incubation with PE Au, displaying the presence of PE Au (arrows), confirming the internalization into the amphisomes (A). (**b**) Equivalent sphere diameter distribution of PVC Au (green) and RAW 264.7 cells, incubated with PVC Au for 48 h (black) at a concentration of 100 µg mL^−1^. Both the SEM (**b_1_**,**b_2_**) and TEM (**b_3_**,**b_4_**) analyses of the sample display the presence of nanoplastics, both internalized and interacting with the cell membrane or as a background material. Particles are visible on the cell membrane and inside the endosome (E), also confirming the presence of Au NPs inside PVC Au by TEM (arrows).

**Figure 6 nanomaterials-13-00594-f006:**
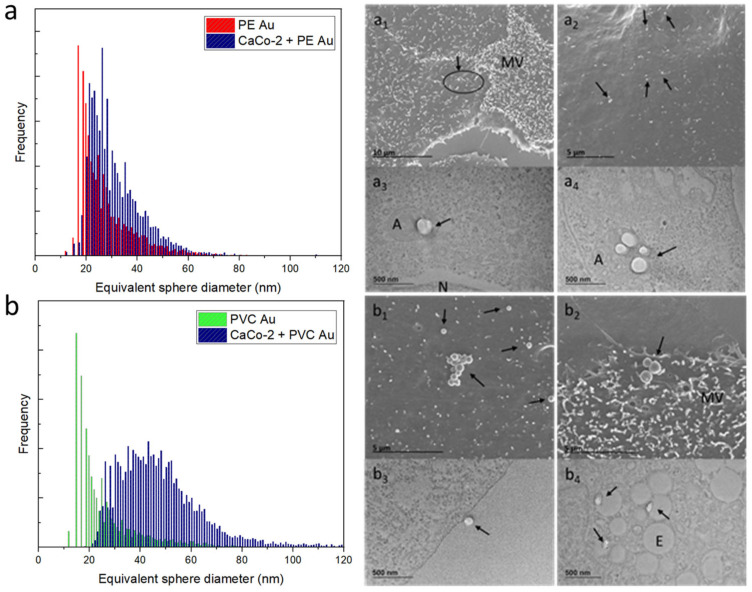
(**a**) Equivalent sphere diameter distribution of PE Au (red) and CaCo-2 cells, incubated with PE Au for 48 h (black) at a concentration of 100 µg mL^−1^. (**a_1_**_,_**a_2_**) SEM images of CaCo-2 cells incubated with PE Au for 48 h reveals the presence of nanoplastics both interacting with the cell membrane and microvilli (MV) (arrows). (**a_3_**,**a_4_**) The TEM images of slices of CaCo-2 cells after 48 h incubation with PE Au, displaying the presence of PE Au (arrows), confirming the internalization in the amphisomes (A); nucleus (N). (**b**) Equivalent sphere diameter distribution of PVC Au and CaCo-2 cells, incubated with PVC Au for 48 h at a concentration of 100 µg mL^−1^. Both SEM (**b_1_**,**b_2_**) and TEM (**b_3_**,**b_4_**) analyses of the sample display the presence of nanoplastics (arrows), both internalized and interacting with the cell membrane. Particles are visible inside the endosome (E) also confirming the presence of Au NPs via TEM.

**Table 1 nanomaterials-13-00594-t001:** A summary of the results obtained by three different detection techniques after the exposure of RAW 264.7 or Caco-2 cells to PE-Au or PVC-Au nanoplastics for 48 h, at concentrations of 1 or 100 µg mL^−1^; (+) indicates the detection of the materials and (-) indicates that no materials are detected.

Cells	Exposure to PE-Au for 48 h					
	1 µg mL^−1^			100 µg mL^−1^		
	scICP-MS	SEM	TEM	scICP-MS	SEM	TEM
RAW 264.7	(+)	(-)	(+)	(+)	(+)	(+)
Caco-2	(+)	(+)	(+)	(+)	(+)	(+)
	**Exposure to PVC-Au for 48 h**					
	**1 µg mL^−1^**			**100 µg mL^−1^**		
	scICP-MS	SEM	TEM	scICP-MS	SEM	TEM
RAW 264.7	(+)	(-)	(+)	(+)	(+)	(+)
Caco-2	(-)	(-)	(-)	(+)	(+)	(+)

scICP-MS: single cell inductively coupled plasma-mass spectrometry; SEM: scanning electron microscope; TEM: transmission electron microscope.

## Data Availability

Raw data are unavailable due to privacy restrictions.

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
