# Peer review of "Investigating the Cellular Uptake of Model Nanoplastics by Single-Cell ICP-MS"

_nanomaterials, 2023, doi:10.3390/nano13030594_

Round 1

Reviewer 1 Report

This work studies the cellular uptake of PE and PVC nanoparticle doped with ultrasmall Au. The interaction of nanopalstics with cells is one the main issues in environmental and nanomaterial research and the use metals as proxies and of single particle and single cell ICPMS as detection technique is one of the trendiest approach right now. Authors demonstrate the capability of single cell ICPMS to provide information about the internalization of nanoplastics by mouse macrophage cells and human intestine cells. The results obtained by single cell ICPMS are compared to the images obtained by TEM.

In summary,  the work proposed by the authors is novel and may attract some interest to the community working with nanoplastics and their interactions in the environment.

However, authors must address several major issues before the manuscript can be accepted for publication:

1)     My first concern is about the title, and more specifically about the use of the word “quantifying”. As I explain below, under the current methodology the quantification of nanoplastics is not feasible. In addition, authors do not show any number (i.e. about the number of nanopalstics internalized). I propose other words like “investigating” but, clearly, in this study authors are not quantifying.

2)     The introduction focuses mainly on microplastics while the target in this study are nanoplastics. I can understand the discussion about why carbon cannot be used to detect nanoplastics by sp-ICP-MS but in recent years there are several studies that have used a similar approach (i.e. to use a metal tag as proxy for detection of particulate plastics sp-ICP-MS) like the ones cited below. This is a new field that is starting to grow and a discussion regarding the current state-of-the-art is mandatory.

Clark, N. J.; Khan, F. R.; Mitrano, D. M.; Boyle, D.; Thompson, R. C., Demonstrating the

translocation of nanoplastics across the fish intestine using palladium-doped polystyrene in a salmon gut-sac. Environment International 2022, 159, 106994.

Jiménez-Lamana, J.; Marigliano, L.; Allouche, J.; Grassl, B.; Szpunar, J.; Reynaud, S., A novel

strategy for the detection and quantification of nanoplastics by single particle inductively coupled plasma mass spectrometry (ICP-MS). Analytical Chemistry 2020, 92, (17), 11664-11672.

 Lai, Y.; Dong, L.; Li, Q.; Li, P.; Hao, Z.; Yu, S.; Liu, J., Counting nanoplastics in environmental waters by single particle inductively coupled plasma mass spectroscopy after cloud-point extraction and in situ labeling of gold nanoparticles. Environmental Science & Technology 2021, 55, (8), 4783-4791.

 Marigliano, L.; Grassl, B.; Szpunar, J.; Reynaud, S.; Jiménez-Lamana, J., Nanoplastic labelling

with metal probes: analytical strategies for their sensitive detection and quantification by icp mass spectrometry. Molecules 2021, 26, (23), 7093.

3)      Authors claim that they can “trace nanoplastics with a size down to 50 nm while maintaining a low doping level of ultrasmall gold nanoparticles” (lines 72-73) and later that the approach prevents “significant alterations of the physical and chemical properties of nanoplastics”. I must totally disagree. In order to trace a nanoplastics particle it may contain enough mass of Au to be seen by single particle ICPMS. Taking into account the instrument used I guess that the limit of detection (in terms of equivalent diameter) must be around 15-20 nm (by the way, this size limit of detection must be calculated and shown in the manuscript either as equivalent diameter or as Au mass or both) which implies that a nanoplastic of 50 nm must contain “minimum” 1/3 of Au in order to be seen. Taking into account the high density of Au it’s difficult to believe that that the nanoplastic will keep their density for example. This must be discussed in the manuscript.   

4)     Related with the previous point and taking a closer look to the distributions shown in Figure 2 is it obvious the distributions are truncated and hence incomplete. This implies that authors are not detecting the small nanoplastics (those that contain a mass of Au below the limit of detection) and hence a quantification of nanoplastics it’s not possible. That’s why it’s important, in order to fully comprehend the study, to show the limit of detection and the limitations of the approach, i.e. it is not quantitative. It would very valuable information to know how many cells have incorporated nanoplastics.

5)     Another critical point when measuring nanoparticles at sizes (or masses) close to the limit of detection is the threshold use in order to differentiate the signal coming from a particle from the background. There are different approaches to calculate this critical limit. However, there is no single mention about this in the manuscript.   

6)     Why the concentration of nanoplastics was chosen at 1 and 100 ppm? Authors claim that this is a compromise “between the detection efficacy of spICP-MS (..)”. However the limiting factor here is the number of cells introduced…

7)     Section 2.3 heading is “sample preparation for sp-ICPMS” while in the text they inly describe the protocol for scICPMS. For the sake of clarity, authors must write one section for spICPMS and other to scICPMS with the parameters used for each technique (for example, what was the TE obtained in spICPMS).

8)     I agree with the authors that the shift toward bigger equivalent diameters (Figures 3a and 3b) implies that cells have internalized more than one nanoplastic. It would interesting to know the average number of nanoplastics internalized per cell (this can be calculated by using the mass distribution instead of the equivalent diameter distribution). Could the authors add more discussion on this point?

9)     In the general discussion about the internalization of nanoplastics in different cell models and at different concentration authors must also take into account that (if I understood well) that they are comparing equivalent diameter distributions obtained by spICPMS (for PE Au and PVC Au) and equivalent diameter distributions obtained by scICPMS (RAW and CaCo-2 + PE/PVC Au). In scICPMS the introduction sample flow (and the gas flow) is way lower than in spICPMS resulting in a significant decrease in sensitivity and hence a higher limit of detection. In the practice, this means that it will be extremely difficult to detect equivalent diameters lower than 20 nm. Therefore, the comparison between distributions can be misleading. For instance, one can think that small nanoplastics has not been internalized but the reality is that technique can’t detect them (the best example is figure 3b). In order to have a proper comparison in the same conditions PE Au and PVC Au must be analyzed by scICPMS. Do the authors have the equivalent diameter distributions of both suspension of nanoplastics obtained by scICPMS? If not, more discussion about this issue must be added in the manuscript   

Author Response

Dear Reviewer,

please find here attached our answer.

'Please see the attachment'

Best regards

Reviewer 2 Report

Authors present in this paper a very interesting study about interaction of cell and nanoplastic with Au particle. More than a study, the paper is a protocol to study this interactions, which is highly timely relevant.

I have absolutely no objections against its publication in Nanomaterials, except two very minor detail:

1) I would like to know the origin of the pristine PE and PVC used in the experiment.

2) Why gold and not another metal. This could be justify or more clearly explained already in the abstract.

Author Response

(The authors gave the same response as above.)

Round 2

Reviewer 1 Report

The author's have addressed correctly all the concerns raised. The manuscript can be published in its current form